# Thermal Conductivities of Choline Chloride-Based Deep Eutectic Solvents and Their Mixtures with Water: Measurement and Estimation

**DOI:** 10.3390/molecules25173816

**Published:** 2020-08-21

**Authors:** Taleb H. Ibrahim, Muhammad A. Sabri, Nabil Abdel Jabbar, Paul Nancarrow, Farouq S. Mjalli, Inas AlNashef

**Affiliations:** 1Department of Chemical Engineering, American University of Sharjah, Sharjah P.O. Box 26666, UAE; ashrafsabri87@gmail.com (M.A.S.); nabdeljabbar@aus.edu (N.A.J.); pnancarrow@aus.edu (P.N.); 2Department of Petroleum and Chemical Engineering, Sultan Qaboos University, P.O. Box 123 Muscat, Oman; farouqsabri@yahoo.com; 3Department of Chemical Engineering, Khalifa University, Abu Dhabi P.O. Box 127788, UAE; enas.nashef@ku.ac.ae

**Keywords:** thermal conductivity, deep eutectic solvent, choline chloride, ionic liquids, group contribution

## Abstract

The thermal conductivities of selected deep eutectic solvents (DESs) were determined using the modified transient plane source (MTPS) method over the temperature range from 295 K to 363 K at atmospheric pressure. The results were found to range from 0.198 W·m^−1^·K^−1^ to 0.250 W·m^−1^·K^−1^. Various empirical and thermodynamic correlations present in literature, including the group contribution method and mixing correlations, were used to model the thermal conductivities of these DES at different temperatures. The predictions of these correlations were compared and consolidated with the reported experimental values. In addition, the thermal conductivities of DES mixtures with water over a wide range of compositions at 298 K and atmospheric pressure were measured. The standard uncertainty in thermal conductivity was estimated to be less than ± 0.001 W·m^−1^·K^−1^ and ± 0.05 K in temperature. The results indicated that DES have significant potential for use as heat transfer fluids.

## 1. Introduction

Over the past two decades, research in the field of ionic liquids (ILs) has grown exponentially due to the favorable properties of many ILs such as: excellent solubility for a wide range of solutes, extremely low volatility, thermal stability, and wide liquid range [1,2]. This research has led to a wide range of potential applications, such as replacements for volatile organic compounds (VOC) industrial solvents [3], catalysis [4], heat transfer fluids and other energy applications [5]. However, while some of these developments have resulted in industrial use, such as in gas processing [6], catalysis [7], and even applications in space technology [8], their uptake on a commercial scale has been inhibited by specific barriers. The first major barrier is their prohibitively high cost, which often limits their use to niche, high value-added applications. The second is the relative shortage of physiochemical and thermal property data which are necessary for preliminary design of industrial scale processes.

More recently, several researchers have moved their attention to a new class of ILs analogues known as deep eutectic solvents (DES). While ILs typically consist of a single ionic compound consisting of one cation and one anion, DESs are formed by mixing hydrogen bond donor (HBD) and hydrogen bond acceptor (HBA) to form a eutectic mixture that has a melting point lower than that of its constituents [9,10]. Although DESs exhibit most of the advantages of ILs, including low melting points, wide liquid ranges, high thermal stability, low volatility, and designable physical and chemical properties, they tend to be significantly less expensive, less toxic, and easier to produce compared to typical ILs. For these reasons, DESs have recently been studied for a wide-range applications in areas such as chemical reactions, catalysis, solvent extraction, nanotechnology, corrosion inhibition, and lubrication [11,12,13,14,15,16,17,18]. However, their potential for use as heat transfer liquids has not been well studied [16,17,18,19,20]. The wise choice of HBA and HBD can give DESs that have high thermal stabilities and low volatilities that make them good potential candidates for heat transfer applications. However, currently, there is a lack of thermal property data for DESs in the literature [20], particularly thermal conductivity (λ) data. Investigation of the thermal properties of DESs is crucial for the coherent design and selection of these DES as heat transfer fluids and process solvents. Yan et al. reported the thermal conductivity of four types of DESs using methyl triphenyl-phosphonium-bromide (MTPB) and choline chloride (ChCl) as HBA, and ethylene glycol (EG) and triethylene glycol (TEG) as HBD from 25 °C to 50 °C [21]. Kucan and Rogošic measured the thermal conductivity of choline chloride-glycerol-based DESs of different molar ratios (ChCl-Gly 1:1.5, 1:2 and 1:3) at 25 °C [22]. Gautam and Seth [23] reported the thermal conductivity of ammonium-based DESs. Choline chloride and N,N-diethyl ethanol ammonium chloride were used as HBAs and urea and N,N-diethylthiourea as HBDs. The thermal conductivities of the DESs have been measured over the temperature range from 298 K to 343 K at atmospheric pressure. They have observed that the thermal conductivities of DESs slightly decreased with increase in temperature. The thermal conductivities of Reline and the DES prepared from urea and N,N-diethyl ethanol ammonium chloride were found to be higher than that of some ILs reported in the literature. Liu et al. measured thermal conductivities of choline chloride/glycerol DES and its TiO_2_, Al_2_O_3_, and graphene oxide-based nanofluids [24]. They found that the thermal conductivity of the developed nanofluids increased by 3–11.4%. Yan et al. reported the thermal conductivity of 33 different DESs derived from phosphonium halide salt and ammonium halide salts and their carbon nanotubes (CNTs) with different concentrations (0.01–0.08 wt%) and at different molar ratio of DESs, CNT concentration, and temperature [21]. Fang et al. investigated the thermal conductivity of functionalized graphene oxide nanoparticles (GNPs) in ammonium- and phosphonium-based DESs without the aid of a surfactant [25]. Different molar ratios of HBAs and HBDs were used to synthesize DESs for the preparation of different concentrations of graphene nanofluids (GNFs). The authors reported that the highest thermal conductivity enhancement of 177% was observed [25]. Dehury et al. reported the thermal conductivity measurements of DESs and alumina-based nanoparticle-dispersed DESs for its use as a potential solar energy storage medium. The authors used different HBDs, e.g., oleic acid, and HBAs, e.g., DL-menthol, for preparing different DESs with different ratios of HBA:HBD [26]. In another study, Singh et al. reported the thermal conductivity values for choline chloride and glycerol or glycerol and polyethylene glycol 600 (PEG) as HBD [27]. The glycerol-based DESs were formed with ratio of 1:2, 1:3, 1:4, and 1:5, whereas three-component DESs of ChCl, PEG, and glycerol were prepared with ratio 1:3:2, 1:4:2, and 1:5:2. The authors found that the thermal conductivity of glycerol-based DESs was less than that of pure glycerol, whereas PEG-based DESs had higher thermal conductivity at temperatures below 60 °C compared to PEG [27]. Dai et al. [28] investigated the dilution effect on the structures and physicochemical properties of natural deep eutectic solvents (NADES) and their improvements of applications. The analytical methods used showed that there were intensive H-bonding interactions between the two components of NADES and that the dilution with water caused the interactions to weaken gradually and eventually disappeared completely at around 50% (*v*/*v*) water. In addition, the authors found that a small amount of water could reduce the viscosity of NADES dramatically and increase the electrical conductivity by up to two order of magnitude for some NADES.

Furthermore, due to the large number of potential DESs that can be synthesized, and the relatively high cost of experimentally determining these properties, the development of accurate and widely applicable estimation methods is necessary [29].

## 2. Modeling of DES Thermal Conductivity

While several efforts have been made to explain the thermal conductivities of liquids via theoretical equations, the results can only be considered qualitatively [30]. Most of these methods are based on the assumption that heat conduction in liquids occurs via longitudinal oscillations and that the thermal conductivity decreases with increasing the average distance between the molecular centers [30]. With regards to ILs in particular, empirically, it has been found that the thermal conductivity decreases slightly with temperature (*T*), and the relationship can be well approximated by a straight line [30,31]:(1)λ=A+BT
where *A* and *B* are empirical constants which can be fitted to experimental data for a given compound. While this approach has a very high accuracy, it requires significant experimental thermal conductivity data for every IL to enable the determination of the empirical constants. Due to the limited availability of thermal conductivity data for ILs and DESs, some researchers have tried to develop methods to estimate thermal conductivity based on other physiochemical properties for which the data are more readily available. Since thermal conductivity is a transport property, attempts have been made to correlate it with another transport property, viscosity (η). The following correlation (Equation (2)), based on Mohanty theory, has been proposed [32,33].
(2)$$\log\left[\frac{\lambda M}{\eta }\right]=1.9596-4.499\times 10^{-3}M$$
where M is the molar mass of IL. To achieve a good fit with the thermal conductivity for a range of alkanes, the correlation assumes the molar mass of IL to be twice the actual value. However, this correlation was tested only against data for four different ILs and, therefore, it is not known if this is reliable over a wide range of ILs and DESs. Froba et al. [34] proposed a correlation based on IL density (ρ) and molar mass (Equation (3)) for thermal conductivity estimation based on experimental data for 10 ILs at atmospheric pressure and 293.15 K.
(3)λMρ=CM+D
where *ρ* is the density (g/cm^3^) of the IL, and *C* and *D* are fitted parameters having values of 0.1130 g·cm^−3^·W·m^−1^·K^−1^ and 22.65 g^2^·cm^−3^·W·m^−1^·K^−1^·mol^−1^, respectively. These parameters were obtained when fitting against data for 36 ILs, resulting in an average absolute relative deviation (AARD) of 6.5%. Wu et al. [35] later tested this model against a wider range of ILs and temperatures, resulting in an AARD of 8.15%.

Gardas et al. [36] developed a simple group contribution model for the prediction of thermal conductivity of ILs as a function of temperature, based on the linear relationship shown in Equation (1). In this approach, the parameters *A* and *B* are determined as a function of the IL structure using group contribution parameters calculated for anions, cation core groups, and hydrocarbon chain groups. The parameters were fitted against the full data set of 107 points for 17 ILs, with an average absolute relative deviation of 1.06%. This further indicates that a linear model is suitable for describing the relationship between temperature and thermal conductivity for ILs, and that the group contribution approach is valid in describing the relationship between IL structure and thermal conductivity. However, without a significant expansion in the group contribution parameters, the model is limited by the relatively small number of ILs for which it can be applied. Furthermore, the training set and test set of data were identical in this study; this means that the model must be considered correlative rather than predictive in nature until tested against a broader test data set.

Hezave et al. [37] used the neural network approach to predict the thermal conductivities for 209 data points from 21 different ILs. The average absolute relative deviation in this case was 0.5%. The “black box” nature of neural networks means that it is not possible to know the relationships between the input and output variables. Therefore, it would not be advisable to use such an approach for prediction beyond the dataset used.

Building upon earlier work by Riedel [38] and Nagvekar and Daubert [39] for organic liquids, Wu et al. [35] proposed the following equation to relate the thermal conductivity of ILs to the reduced temperature, *T_r_*:(4)$$\lambda =\lambda_{0}\left[1+k_{0}{\left(1-T_{r}\right)}^{\frac{2}{3}}\;+1\right]$$
where *k_0_* is the temperature-independent constant, and the reduced temperature is calculated from *T_r_* = *T*/*T_c_*, where *T_c_* is the critical temperature for the IL. Although the critical temperatures for ILs cannot be determined experimentally, Valderrama et al. [40] used a group contribution method to estimate critical properties of ILs from their chemical structure. $\lambda_{0}$ is determined via the following group contribution method:(5)$$\lambda_{0}=\overset{2}{\underset{i=0}{\sum }}a_{i}{\left[\overset{k}{\underset{j=1}{\sum }}n_{j}∆\lambda_{0,j}\right]}^{i}$$
where *n_j_* is the number of functional groups of type *j*, *k* is the number of different functional groups in the species, and *a_i_* and $∆\lambda_{0}$ are parameters obtained by fitting the model to experimental data.

The model was concluded to have an average absolute deviations of 1.66%; however, deviations such as 5.88% have also been reported for some ILs [35]. Nevertheless, this model predicts thermal conductivity over a wide range of temperatures for various ILs with sufficient accuracy for most applications and, therefore, it is considered a good basis for predicting DESs thermal conductivity.

To enable the application of the Wu model for thermal conductivity estimation of DES in this work, the critical temperature and normal boiling points of the pure precursors were estimated using the Valderrama group contribution method as mentioned below [40].
(6)$$T_{B}=198.2+\;\overset{m}{\underset{i=1}{\sum }}n_{i}∆T_{Bi}$$
(7)$$T_{c}=\;\frac{T_{B}}{0.5703+1.0121\sum_{i=1}^{m}n_{i}∆T_{Ci}-{\left[\sum_{i=1}^{m}n_{i}∆T_{Ci}\right]}^{2}}$$
where *T_B_*, *T_C_*, ∆*T_Bi_*, and ∆*T_Ci_* are the boiling temperature, critical temperature, group contribution parameter for boiling point, and group contribution parameter for critical temperature, in Kelvin, respectively. Using these properties, the thermal conductivities of pure precursors at the temperatures of interest were obtained by implementing the Wu model (Equation (4)) [35].

Since a DES is basically a mixture of two or more precursors, the final properties can be estimated using the Lee-Kesler mixing rules or similar correlations. To estimate the final critical temperatures and boiling points for the DESs studied from those for their constituents, the Knapp et al. recommendations were employed [41,42]. The DESs under consideration are binary (salt + HBD) in nature; thus, the thermal conductivities of the DESs were estimated using the thermal conductivity correlations for binary mixtures, such as the Jameison and Fillipov correlations [43], as represented by Equations (8) and (9), respectively:(8)$$\lambda_{m}=w_{1}\lambda_{1}+w_{2}\lambda_{2}-\alpha \left(\lambda_{2}-\lambda_{1}\right)\left[1-{\left(w_{2}\right)}^{\frac{1}{2}}\right]w_{2}$$
(9)$$\lambda_{m}=w_{1}\lambda_{1}+w_{2}\lambda_{2}-\alpha w_{1}w_{2}\left(\lambda_{2}-\lambda_{1}\right)$$
where λ*_m_* represents the thermal conductivity of the mixture, *w*_1_ and *w*_2_ are the mass fractions of the component 1 and component 2, and λ_1_ and λ_2_ are the thermal conductivities of the pure component 1 and pure component 2. The components are chosen such that λ_2_ ≥ λ_1_ for Jameison and Filippov correlations.

Herein, the thermal conductivities of seven choline chloride-based DESs: reline, tegaline, maline, glyceline, ethaline, glucoline, and fructoline, over the temperature range from 298 K to 363 K, are reported. Due to the hydrophilic nature of these DESs, the effect of water content on thermal conductivity has also been measured over the full compositional range. Furthermore, a methodology for the prediction of DES thermal conductivity, created by the combination of the Valderrama group contribution method, Wu model, and Jameison or Fillipov correlation, collectively described herein as the Extended Wu Model (EWM), is developed and fitted against the experimental data.

## 3. Results and Discussion

The thermal conductivities of seven DES were measured at 298 K and are reported in Table 1. The results are relatively similar, ranging from 0.1978 W·m^−1^·K^−1^ to 0.2410 W·m^−1^·K^−1^. The following trend was observed, arranged from highest to lowest thermal conductivity: Reline > glyceline > glucoline > fructoline > ethaline > tegaline > maline.

It is worth noting that the values of thermal conductivity for Reline in this work are in excellent agreement with those reported by Gautam and Seth [23]. In addition, the thermal conductivity of glyceline at 298 K, 0.223 W·m^−1^·K^−1^, is close enough to that reported by Kucan and Rogošic [22], 0.232. Moreover, there is less than 5% difference between the thermal conductivity of tegaline in this work and that reported by Yan et al. [21].

The thermal conductivity of ILs depends on several factors, including the cation and anion of IL, structure of the IL, viscosity and molecular weight. Since DESs are closely related to classical ILs in terms of chemical structure, it is expected that the thermal conductivities of DESs also depend on these factors. Each of the DESs studied here contains choline chloride as the HBA; hence, the difference in thermal conductivity of DES is attributed to the chemical structure of HBD. It is observed that, for compounds with similar structure, as the molecular weight of the HBD increases, the thermal conductivity of the respective DES decreases. Urea, glycerol, and sugars (d-fructose and d-glucose) have an inverse relationship of molecular weight to the thermal conductivities of their respective DES. Ethylene glycol and triethylene glycol-based DES have similar thermal conductivity, but they also follow the same trend. Owing to the similar molecular weight, but different structure, fructoline and glucoline thermal conductivities almost overlap each other. These observations are in agreement with the theory for molecular liquids, described previously, that thermal conductivity decreases as the distance between molecular centers increases.

Figure 1 shows the results obtained for the measurement of the thermal conductivity versus temperature for the DESs investigated in this work as a function of temperature.

It is clear from Figure 1 that, within the studied range, the effect of temperature on the thermal conductivities of the DESs is relatively weak and follows a linear trend, with thermal conductivity decreasing as temperature increases. The same behavior for different DESs was reported by Gautam and Seth [23]. Previous studies on conventional ILs also demonstrated a similar linear relationship, decreasing with temperature [30,34,44]. This could be attributed to the inverse relationship between DES density and temperature [45], which results in the ions or molecules moving further apart as temperature increases, the resulting decrease in thermal conductivity is expected.

The linear model has been applied to the data, as shown in Figure 1, and the corresponding fitted parameters are shown in Table 2. The linear model clearly gives a good fit to the experimental data, as indicated by the high *R*^2^ values, except for glyceline and reline, Table 3.

To extend the Wu group contribution model for IL thermal conductivity towards the estimation of DES thermal conductivity, firstly, the critical properties of DES were estimated using the Valderrama method mentioned previously. The calculated values are listed in Table 3. All the calculated critical values are in good match with previously reported values [41,46]. The values for tegaline and maline, for the composition under consideration, have not been previously reported in literature.

Table 4 shows the results obtained by fitting two correlations, Filippov and Jamieson, to the thermal conductivity versus temperature data for all the DESs studied in this work.

As observed, both correlations correctly describe the temperature dependence of the thermal conductivity for the DESs, with similar percentage errors.

Next, the methodology was extended from a correlative approach to a predictive approach by incorporating the Wu group contribution model. The parameter λ*_0_* was calculated from the functional group parameters for each component in the DES according to the methodology outlined in the literature [35].

EWM shows a good fit, though not as good as the linear fit, with an average absolute deviation of less than 1.75% and a maximum absolute deviation of less than 3.5% in each case. Errors in this range are within the expected experimental error. Thus, the provided method can be used effectively for thermal conductivity determinations; however, care must be taken to determine the group contributions and the type of mixing rule applied. The only major limitation in the applicability of this method is the data availability for ∆λ values of groups present in the molecules making the DES. The final results of the model estimations with average absolute errors are given in Table 5 and are shown in Figure 2.

### Thermal Conductivity of DES-Water Mixtures

Due to the mutual miscibility of water and the studied DESs, and the likelihood that they may exist in aqueous mixtures in many potential applications, the thermal conductivity of DES-water mixtures were also investigated at 298 K with weight fractions of DES ranging from 0 to 1. The results are shown in Figure 3 and the detailed data is reported in Table 5. Since the thermal conductivity of water is significantly higher than that for each DES, as expected, the thermal conductivity decreases as the weight fraction of DES increases. However, the relationship is not linear. It is clear that the thermal conductivity of each mixture is less than the weighted average of the components, an observation which is common in previous studies for IL-water mixtures [30]. It is worth mentioning that as the fraction of water increases, the DES loses its characteristics because of the hydrogen bonding between the water and both components of the DES. When the fraction reaches a certain value, the DES will lose all its properties, and the result is an aqueous solution of the two components of the DES with complex hydrogen bonding among the three components [47,48].

The DES-water mixture results were correlated using three different models: Jameison, Filippov, and Rowley. In general, the Filippov correlation provides the most satisfactory fit for the experimental data. Table 6 provides the results for the three correlations and the value of the related parameters for each case, respectively. Figure 3 depicts the experimental values of thermal conductivities of DES-water binary solutions and the fitted model values using Filippov correlation.

For glucoline and fructoline, the Jameison correlation with values of α = 0.9303 and 0.9422 provides the best fit with a maximum absolute error (MAE) of 1.37%. The maximum absolute errors observed in each case for the studied correlations were summarized in Table 7. As observed from the maximum absolute errors, the Rowley correlation yields larger errors than either Jamieson or Filippov. The Filippov correlation seems to be more suitable in these cases since in each case, the maximum error is near 2 percent, which is within the experimental error.

The variation in thermal conductivity values of different DES can be attributed to the type of salt and HBD, their respective compositions, and the intermolecular interactions between them. The thermal conductivity of DES-water mixtures is dependent mainly upon their mass fractions within the binary mixture as well as the chemical nature of the DES and its interaction with water. Parameters such as viscosity, molecular weight, and critical properties change as the mass fraction of the DES in the binary mixture changes, which in turn influences the thermal conductivity of the mixture. The thermal conductivity of DES is quite low compared to water and any slight variation in the composition increases the deviations and possible errors in the mixture calculated thermal conductivity values. In an ideal mixture, thermal conductivity is expected be the weighted average of the thermal conductivity of the components; however, it is clear that the aqueous solutions of all studied DESs showed non-ideal behavior. This may be attributed to the interaction forces among the constituents in the solution, mainly, hydrogen bonding. Aqueous solutions of glucoline and fructoline showed similar deviations from their weighted average, which is due to their similar chemical structure and molecular weight. High deviations were observed in the cases of glucoline, fructoline, ethaline, and reline, while mild deviations were observed in the case of glyceline (Figure 4).

## 4. Materials and Instrumentation

Chemically pure anhydrous glycerol, d-glucose, ethylene glycol, malonic acid (99% min), and urea (99% min) were purchased from LabChem Inc, Zelienople, PA, USA. D-fructose (extra pure) and triethylene glycol (extra pure) was supplied by SDFCL, Mumbai, India and Schartau Chemie S.A., Barcelona, Spain, respectively. A TCi analyzer (TH91-13-00631, C-Therm Technologies Ltd., Fredericton, NB, Canada) was used to measure the thermal conductivity in combination with double-distilled water (Water Still Aquatron A4000D, stuart equipement, Staffordshire, UK). A precise vacuum oven (Model WOV-30, DAIHAN Scientific Co. Ltd., Wonju-Si, Gangwon-do, Korea) fitted with a vacuum pump (Model G-50DA, UlvacKiko, Saito-City, Miyazaki, Japan) was used for drying the DESs after preparation using a hot plate stirrer (MSH-20D, DAIHAN Scientific Co. Ltd., Wonju-Si, Korea).

### 4.1. DES Preparation

21.00 g of the HBA, choline chloride, was mixed with 13.55 g d-fructose, 13.55 g d-glucose, 67.76 g triethylene glycol, 42.47 g phenol, 27.70 g glycerol, 15.65 g malonic acid, 18.07 g urea, and 18.67 g ethylene glycol, respectively, according to their respective molar ratios, as mentioned in literature and summarized in Table 7. In each case, the mixture of HBA and related hydrogen bond donor (HBD) was shaken at 400 rpm and 353 K for two hours to produce a stable DES with no apparent precipitation. All prepared DESs were dried under vacuum for 6 h at 333 K prior to use and were found to have a water content of less than 0.1% by mass as determined by Karl Fischer titration.

### 4.2. Thermal Conductivity Measurements

A TCi thermal conductivity analyzer (C-Therm Technologies Ltd.) was utilized for measuring the thermal conductivity of the neat and aqueous solutions of DESs via the modified transient plane method [52]. This approach has been reliably used in various previous studies on both solids and liquids [52,53,54,55,56]. The TCi system consists of a spiral-shaped sensor surrounded by a guard ring approximating one-dimensional heat flow to the material from the sensor, causing a rapid voltage drop across the heating source, which allows the emissivity and thermal conductivity of the sample under consideration to be measured. The guard ring allows unidirectional heat flux from the material to the sensor, thus cancelling any thermal edge effect. Further details and theoretical background of TCi thermal conductivity analyzer has been presented by Adam et al. [52].

Five readings of thermal conductivity for each sample under consideration were taken and found to be highly consistent. The average of the five readings is reported in this work. The standard uncertainty was found to be ± 0.001 W·m^−1^·K^−1^ and ± 0.05 K for thermal conductivity and temperature measurements, respectively.

## 5. Conclusions

The thermal conductivities of selected choline chloride-based DESs have been reported for the temperature range of 295 K to 363 K at ambient pressure. Generally, the measured DES thermal conductivities are similar to those reported for ILs. Thermal conductivity was found to be strongly related to the molecular structure of the DES constituents. The thermal conductivity decreased slightly with the increase of temperature. The thermal conductivities of these DESs were found to be less than that the weighted average of respective pure components. This can be attributed to the strong interactions between the constituents of the DES. A group contribution method was applied in conjunction with other models to predict the thermal conductivity of DESs at wide range of temperatures. The model predictions described in this work were compared with the experimental values. A good agreement with experimental results was observed with an average absolute deviation of less than 1.75%. This study paves the road to conduct further thermal conductivity studies for other types of DES and further extension of the proposed predictive method.

## Figures and Tables

**Figure 1 molecules-25-03816-f001:**
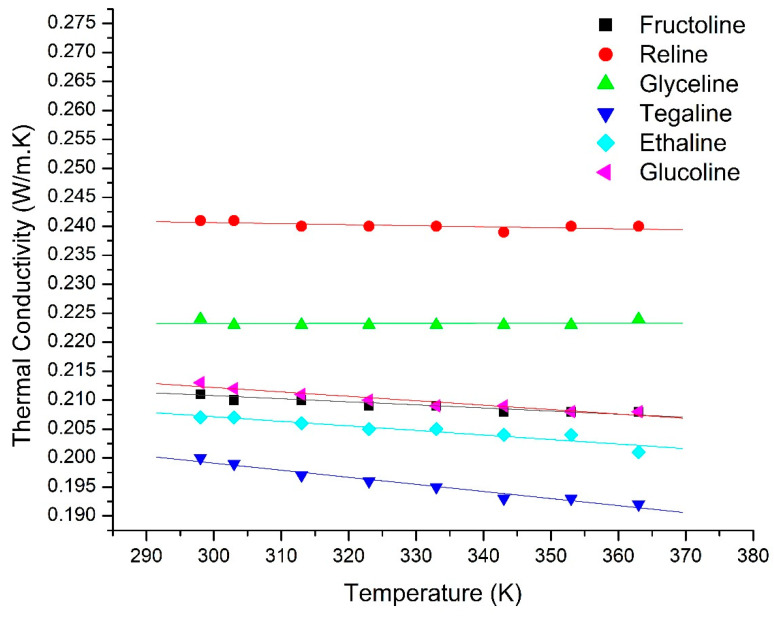
Thermal conductivity of different DESs over the temperature range from 295 K to 363 K. The dots are experimental values while the solid lines represent the fitted linear model shown in Equation (1).

**Figure 2 molecules-25-03816-f002:**
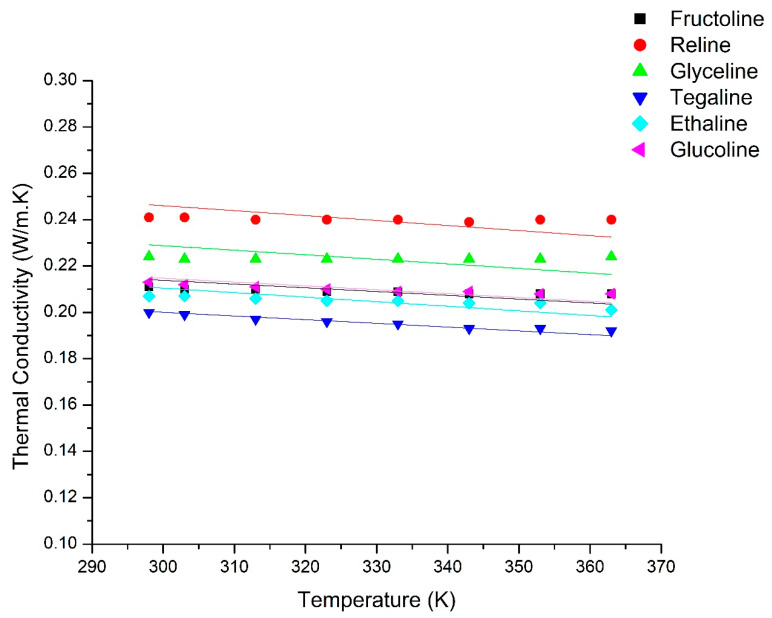
Thermal conductivity versus temperature calculated using EWM approach (lines) compared with the experimental data (points).

**Figure 3 molecules-25-03816-f003:**
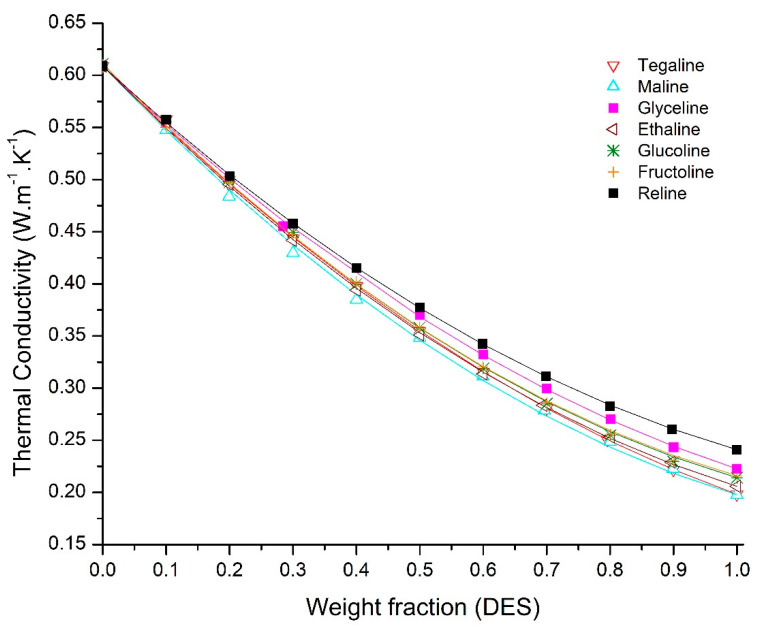
Thermal conductivities of DES aqueous solutions as a function of DES mass fraction. Solid lines represent Filippov model correlations.

**Figure 4 molecules-25-03816-f004:**
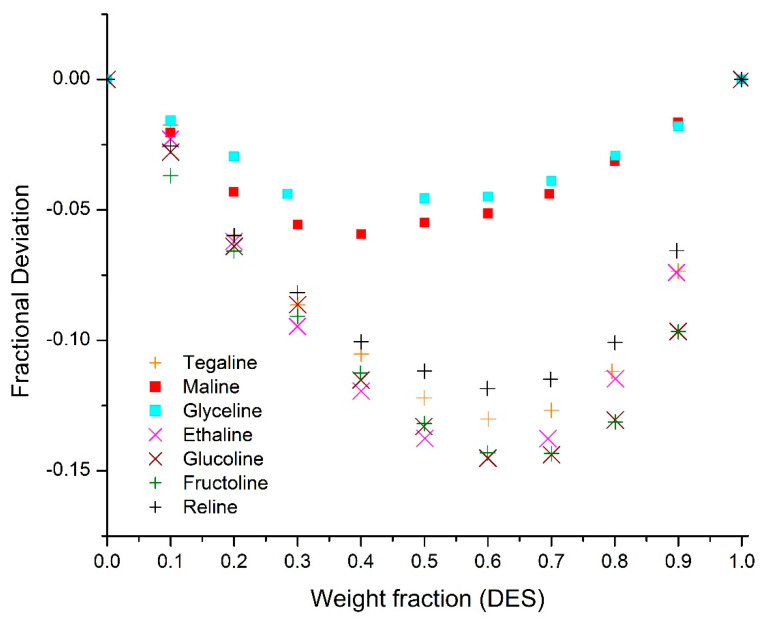
Thermal conductivity deviations between DESs aqueous solutions and their ideal mixtures. *based on temperature range 313 K to 353 K.

**Table 1 molecules-25-03816-t001:** Thermal conductivities of neat DESs at 298 K.

DES	$\;\lambda \;(W\text{·}m^{-1}\text{·}K^{-1})$
Fructoline	0.211
Glucoline	0.214
Tegaline	0.198
Ethaline	0.206
Glyceline	0.223
Maline	0.198
Reline	0.241

**Table 2 molecules-25-03816-t002:** Fitting parameters, *A* and *B*, for Equation (1) relating thermal conductivity with temperature for selected DES.

DES *	*A*(W·m^−^^1^·K^−^^1^)	*B*(W·m^−1^·K^−2^)	*R* ^2^
Tegaline	0.23578	−1.22195 × 10 ^−4^	0.957
Glucoline	0.23504	−7.61905 × 10 ^−5^	0.962
Glyceline	0.22293	9.68523 × 10 ^−7^	0.761
Fructoline	0.22676	−5.32688 × 10 ^−5^	0.933
Reline	0.24612	−1.82405 × 10 ^−5^	0.704
Ethaline	0.23071	−7.86118 × 10 ^−5^	0.839

* Based on temperature range 313 K to 353 K.

**Table 3 molecules-25-03816-t003:** Critical properties of DES estimated using the Valderrama method **[40]**.

DES	*T_c_* (K)	*V_c_* (Units)
Reline	641.89	254.37
Tegaline	712.69	441.22
Maline	707.33	348.64
Glyceline	680.67	315.17
Ethaline	602.00	259.67
Glucoline	757.44	449.80
Fructoline	756.99	453.54

**Table 4 molecules-25-03816-t004:** Average absolute and maximum absolute deviations for EWM for thermal conductivity estimation over within temperature range of 298 K to 363 K for selected DESs.

DES	Jameison	Filippov	Average Absolute Deviation (%)	Maximum Absolute Deviation (%)
	A	α		
Reline	1.451	0.01745	1.54	2.97
Glucoline	0.159	0.17436	0.82	1.63
Fructoline	−0.343	−0.66909	1.08	2.23
Tegaline	2.107	0.48537	0.47	1.09
Glyceline	0.923	0.82089	1.64	3.24
Ethaline	1.153	0.95989	1.22	2.13

**Table 5 molecules-25-03816-t005:** Thermal conductivity of aqueous solutions of DESs at varying DES mass fraction (X).

Reline		Tegaline		Maline		Glyceline		Ethaline		Glucoline		Fructoline	
X	λ	X	Λ	X	Λ	X	Λ	X	λ	X	Λ	X	λ
	W·m^−1^·K^−1^		W·m^−1^·K^−1^		W·m^−1^·K^−1^		W·m^−1^·K^−1^		W·m^−1^·K^−1^		W·m^−1^·K^−1^		W·m^−1^·K^−1^
1.0000	0.241	1.0000	0.198	1.0000	0.198	1.0000	0.223	1.0000	0.206	1.0000	0.214	1.0000	0.213
0.8975	0.260	0.8997	0.222	0.8994	0.223	0.9002	0.243	0.8970	0.229	0.8999	0.229	0.9000	0.231
0.8001	0.283	0.7955	0.251	0.8002	0.249	0.8006	0.270	0.8010	0.253	0.8007	0.255	0.8008	0.256
0.6986	0.311	0.7001	0.281	0.6965	0.279	0.7001	0.299	0.6946	0.284	0.7003	0.285	0.7004	0.286
0.5992	0.342	0.6007	0.315	0.5998	0.311	0.6000	0.332	0.5999	0.314	0.6000	0.319	0.5996	0.321
0.5000	0.377	0.4996	0.354	0.4998	0.349	0.4998	0.370	0.5008	0.351	0.4992	0.358	0.4999	0.359
0.4002	0.415	0.4004	0.398	0.4002	0.385	0.4002	0.409	0.4004	0.394	0.3999	0.400	0.3995	0.402
0.2998	0.458	0.3002	0.444	0.3000	0.430	0.2841	0.455	0.2998	0.442	0.3003	0.449	0.3000	0.448
0.2001	0.503	0.1994	0.495	0.1994	0.484	0.2000	0.502	0.2000	0.495	0.2001	0.498	0.2000	0.497
0.1001	0.557	0.1000	0.558	0.0998	0.547	0.1000	0.554	0.0998	0.556	0.1000	0.555	0.1000	0.550
0.0000	0.609	0.0000	0.609	0.0000	0.609	0.0000	0.609	0.0000	0.609	0.0000	0.609	0.0000	0.609

**Table 6 molecules-25-03816-t006:** Fitting parameters for the Jamieson, Filippov, and Rowley models for DES-water binary solutions at different mass ratios.

DES	Jamieson		Filippov		Rowley		
	*A*	Sum of Squared Errors	*α*	Sum of Squared Errors	G_12_	G_21_	Sum of Squared Errors
Reline	0.8735	9.16 × 10^−5^	0.5237	1.08 × 10^−5^	0.3462	3.5866	4.70 × 10^−4^
Tegaline	0.7894	1.35 × 10^−4^	0.4732	5.95 × 10^−5^	0.2945	3.5865	6.54 × 10^−4^
Maline	0.9210	6.31 × 10^−4^	0.5571	2.04 × 10^−4^	0.3918	1.1124	1.39 × 10^−3^
Glyceline	0.8065	1.52 × 10^−4^	0.4905	3.72 × 10^−5^	0.2985	3.5865	6.41 × 10^−4^
Ethaline	0.8933	1.87 × 10^−4^	0.5360	5.91 × 10^−5^	0.3606	3.5866	6.62 × 10^−4^
Glucoline	0.9303	2.60 × 10^−5^	0.5551	6.83 × 10^−5^	0.3766	3.5866	2.69 × 10^−4^
Fructoline	0.9422	4.70 × 10^−5^	0.5630	4.61 × 10^−5^	0.3848	3.5866	3.28 × 10^−4^

**Table 7 molecules-25-03816-t007:** Compositions of the DESs used in this study.

DES	HBA	HBD	HBA:HBDMolar Ratio	Reference
Fructoline	Choline Chloride	d-fructose	2:1	[49]
Glucoline	Choline Chloride	d-glucose	2:1	[20]
Tegaline	Choline Chloride	Triethylene glycol	1:3	[50]
Ethaline	Choline Chloride	Ethylene glycol	1:2	[51]
Glyceline	Choline Chloride	Glycerol	1:2	[51]
Maline	Choline Chloride	Malonic acid	1:1	[51]
Reline	Choline Chloride	Urea	1:2	[51]

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
