# Peer review of "Thermal Conductivities of Choline Chloride-Based Deep Eutectic Solvents and Their Mixtures with Water: Measurement and Estimation"

_molecules, 2020, doi:10.3390/molecules25173816_

Round 1

Reviewer 1 Report

In this contribution, the authors studied the thermal conductivity of a series DES in both anhydrous and hydrated conditions. They built models upon the existing knowledge in this matter for traditional ionic liquids. The manuscript shows precise results that are expected to serve as a starting point for several applications related to DESs as nonvolatile solvents.
I suggest accepting the manuscript after the authors consider the following minor comments and suggestions.
The nomenclature adopted by the authors to name the different DESs may lead to future confusion as to what is the molar ratio that the authors considered the archetypical between the HBD and HBD under study. If the authors instead refer to each DES as HBD-HBA followed by their molar composition, there is no room for errors, e.g., Urea-Choline Chloride 2:1, respectively.
Many of the DES's physicochemical characteristics are already considered by some of the models applied to predict and establish the trends between a given DES and its thermal conductivity. However, I suggest adding, as supporting information, the relationships between each DES's thermal conductivity with critical physical properties like density, viscosity, molecular weight, etc. This would make more visual these relationships and may trigger the search for new relationships that could eventually end up in new models.

Author Response

Dear Editor,

We would like to thank the reviewer for the valuable comments. Please find find below our response for the comments:

In this contribution, the authors studied the thermal conductivity of a series DES in both anhydrous and hydrated conditions. They built models upon the existing knowledge in this matter for traditional ionic liquids. The manuscript shows precise results that are expected to serve as a starting point for several applications related to DESs as nonvolatile solvents. 
I suggest accepting the manuscript after the authors consider the following minor comments and suggestions.
The nomenclature adopted by the authors to name the different DESs may lead to future confusion as to what is the molar ratio that the authors considered the archetypical between the HBD and HBD under study. If the authors instead refer to each DES as HBD-HBA followed by their molar composition, there is no room for errors, e.g., Urea-Choline Chloride 2:1, respectively. 

Response: Table 1 states the nomenclature and the exact molar composition without any room for error. The molar composition ratio of the HBD and HBA are represented clearly in Table 1 column 4.

Many of the DES's physicochemical characteristics are already considered by some of the models applied to predict and establish the trends between a given DES and its thermal conductivity. However, I suggest adding, as supporting information, the relationships between each DES's thermal conductivity with critical physical properties like density, viscosity, molecular weight, etc. This would make more visual these relationships and may trigger the search for new relationships that could eventually end up in new models.

We thank the reviewer for this valuable suggestion, which is extremely important. However, this needs a lot of time and effort. In addition, data from literature must also be used in addition to the data obtained in this work. This will be done in a separate work in the near future.

Reviewer 2 Report

The work titled

 Thermal Conductivities of Choline Chloride-Based Deep Eutectic Solvents and their Mixtures with Water: Measurement and Estimation requires some supplementary  before publication.

Page 1

Line 42 – thermal stability of some DES is disputable

line 79 - correct into Modeling

line 81 - correct into qualitatively

Page 2

Lines 56-75 Thermal conductivity of starch modified with choline chloride/urea DES was measured in work Adamus et al. Thermoplastic starch with deep eutectic solvents and montmorillonite as a base for composite materials. Industrial Crops and Products, 2018, 123, 187-227.

Page 4

Was glycerol anhydrous ?

Page 5 – Discussion – viscosity of DES should be Take into consideration in this study.

Table 1 I suggest change the order of DES in the table, ethaline should be close to tegaline and the same sequence should be in Table 2.

‘molecular weight increasing-TCi decreasing’ - Reline did not confirm your conclusions, The authors should take other properties and mutual interactions between DES components into consideration, eg. viscosity, hydrogen bonding, thermal properties of DES. Choline chl./urea is typical DES, due to Tm, hovewer, choline chl./carboxylic acids are more like LTTMs – they reveals Tg on DSC curves.

Page 7 Reline and Glyceline have low R2, chich can suggest that they do not fit to the model.

Point 4.1 Individual components of DES with water and their thermal conductivity should be presented here for comparison. The same for Glycerol and glycols.

Page 8 line 279  this work is worth to menton too Dai, Y. et al. Tailoring properties of natural deep eutectic solvents with water to facilitate their application. Food Chemistry, 2015, 187, 14–19.

Author Response

We thank the reviewer for his valuable comments. The reviewer comments were considered and is our reply:

Page 1

Line 42 – thermal stability of some DES is disputable

The statement was adjusted accordingly.

line 79 - correct into Modeling

Done

line 81 - correct into qualitatively

Done

Page 2

Lines 56-75 Thermal conductivity of starch modified with choline chloride/urea DES was measured in work Adamus et al. Thermoplastic starch with deep eutectic solvents and montmorillonite as a base for composite materials. Industrial Crops and Products, 2018, 123, 187-227.

With all respect to the Reviewer, we do not think that Adamus et al. work is relevant to the present work.

Page 4

Was glycerol anhydrous ?

Yes. This was added in the Materials and Instrumentation section.

Page 5 – Discussion – viscosity of DES should be Take into consideration in this study.

Table 1 I suggest change the order of DES in the table, ethaline should be close to tegaline and the same sequence should be in Table 2.

Resposne: Done

‘molecular weight increasing-TCi decreasing’ - Reline did not confirm your conclusions, The authors should take other properties and mutual interactions between DES components into consideration, eg. viscosity, hydrogen bonding, thermal properties of DES. Choline chl./urea is typical DES, due to Tm, hovewer, choline chl./carboxylic acids are more like LTTMs – they reveals Tg on DSC curves.

Page 7 Reline and Glyceline have low R2, chich can suggest that they do not fit to the model.

We modified the discussion to indicate that Reline and Glyceline have low R2

Point 4.1 Individual components of DES with water and their thermal conductivity should be presented here for comparison. The same for Glycerol and glycols.

We do agree with the Reviewer regarding this comment, however, due to the current crisis and limited access to the labs, this cannot be done at present. It will be done in the future as a separate work.

Page 8 line 279  this work is worth to menton too Dai, Y. et al. Tailoring properties of natural deep eutectic solvents with water to facilitate their application. Food Chemistry, 2015, 187, 14–19.

The suggested work was referred to in the introduction.

Reviewer 3 Report

The manuscript needs some corrections before being accepted for publication.

  1. The Introduction section is too long and not focused on the subject of the paper and therefore it needs to be shortened and only the relevant aspects to be included.
  2. Throughout the paper some of the symbols used for the structural parameters are not visible (for example page 4, line 137).
  3. The presentation of the number of DESs investigated in this paper is not consistent and rather unclear; p. 4 line 163 mentioned 6 DESs, followed by a list of seven mixtures. p. 4&5, lines 171 and 181 mentioned the use of phenol which is not seen further in the experimental data. The same confusion is maintained further when the data for seven DESs are presented in Table 1 while Figure 1 shows the data just for six. The authors should correct and explain why Table 3 contains data for "six selected DESs" while in Table 4 data for seven mixtures are presented.
  4. Even if in the "Conclusion" section the authors say that the thermal conductivity is strongly related to the molecular structure", this aspect is not clearly explained in the paper.

Author Response

We would like to thank the reviewer for the valuable comments. Below is our reply:

The Introduction section is too long and not focused on the subject of the paper and therefore it needs to be shortened and only the relevant aspects to be included.

The introduction was revised according to the comment of the reviewer.

Throughout the paper some of the symbols used for the structural parameters are not visible (for example page 4, line 137).

The missing symbols were written properly.

The presentation of the number of DESs investigated in this paper is not consistent and rather unclear; p. 4 line 163 mentioned 6 DESs, followed by a list of seven mixtures. p. 4&5, lines 171 and 181 mentioned the use of phenol which is not seen further in the experimental data. The same confusion is maintained further when the data for seven DESs are presented in Table 1 while Figure 1 shows the data just for six. The authors should correct and explain why Table 3 contains data for "six selected DESs" while in Table 4 data for seven mixtures are presented.

Phenol was removed from the text. The number of investigated DESs was seven and not six as was mentioned in parts of the manuscript. In some cases six DESs were tested because bubbles formed and hence the thermal conductivity can’t be measured.

Even if in the "Conclusion" section the authors say that the thermal conductivity is strongly related to the molecular structure", this aspect is not clearly explained in the paper.

The effect of the structure of the DESs constituents was mentioned in the manuscript. However, because choline chloride is common for all DESs, the effect of the structure of the HBD is the main factor that affected the thermal conductivity. For example, the highest thermal conductivity, 0.242 at 25°C was for urea as HBD. It was also mentioned in the discussion that “Aqueous solutions of glucoline and fructoline showed similar deviations from their weighted average, which is due to their similar chemical structure and molecular weight.”

Round 2

Reviewer 2 Report

'Point 4.1 Individual components of DES with water and their thermal conductivity should be presented here for comparison. The same for Glycerol and glycols.

We do agree with the Reviewer regarding this comment, however, due to the current crisis and limited access to the labs, this cannot be done at present. It will be done in the future as a separate work'

- Did Auhors try to find the data in the literature and compare instead of conducting analysis?